# The curious case of the structural phase transition in SnSe insights from neutron total scattering

Bo Jiang [1,2,10], Jennifer Neu [3,4,5,9,10], Daniel Olds [1,6], Simon A. J. Kimber [7], Katharine Page [1,2] ✉ & Theo Siegrist [3,8] ✉

At elevated temperatures SnSe is reported to undergo a structural transition from the low symmetry orthorhombic GeS-type to a higher symmetry orthorhombic TlI-type. Although increasing symmetry should likewise increase lattice thermal conductivity, many experiments on single crystals and polycrystalline materials indicate that this is not the case. Here we present temperature dependent analysis of time-of-flight (TOF) neutron total scattering data in combination with theoretical modeling to probe the local to long-range evolution of the structure. We report that while SnSe is well characterized on average within the high symmetry space group above the transition, over length scales of a few unit cells SnSe remains better characterized in the low symmetry GeS-type space group. Our finding from robust modeling provides further insight into the curious case of a dynamic order-disorder phase transition in SnSe, a model consistent with the soft-phonon picture of the high thermoelectric power above the phase transition.

Tin selenide, SnSe, has been the focus of intense interest as a thermoelectric material, with a low lattice thermal conductivity between 600 K and 800 K, and reciprocally, a large power factor[1–6]. The structural phase transition in SnSe, from orthorhombic GeS-type with *Pnma* symmetry at low temperature to orthorhombic TlI-type with symmetry *Cmcm* at high temperature is at the heart of the low lattice thermal transport coefficient[7–12]. However, the nature of the phase transition remains under investigation: is it a displacive phase transition similar to most ferroelectric materials, or an order-disorder transition with dynamic disorder in the high-temperature phase? The initial identification of the phase transition as displacive, with a well-defined transition temperature following group/sub-group relationships, was based on X-ray and neutron diffraction experiments[13,14]. Though sometimes still discussed as an order-disorder phenomena, the modern consensus built upon optical methods, inelastic neutron scattering, and coupled theoretical probes[12,15–19], is that the phase transition is driven by condensation of a highly anharmonic zone-boundary (Y-point) phonon within the high-temperature phase. In particular, theoretical calculations involving three-body forces have shown a double-well 0 K potential energy surface that leads to intrinsic phonon anharmonicity within the high-temperature regime;[12] and when third-order force constants are determined in a non-perturbative manner, the experimentally-observed phonon instability within the *Cmcm* phase is recovered and explains the low lattice thermal conductivity at high temperatures[16]. Despite strong experimental and theoretical evidence of the soft

[1]Neutron Scattering Division, Oak Ridge National Laboratory, Oak Ridge, TN 37830, USA. [2]Materials Science and Engineering Department, University of Tennessee, Knoxville, TN 37996, USA. [3]National High Magnetic Field Laboratory, Florida State University, Tallahassee, FL 32310, USA. [4]Dept. of Physics 77 Chieftain Way, Florida State University, Tallahassee, FL 32306-4350, USA. [5]Dept. of Chemistry & Biochemistry 95 Chieftain Way 118 DLC, Florida State University, Tallahassee, FL 32306-4390, USA. [6]National Synchrotron Light Source II, Brookhaven National Laboratory, Upton, NY 11973-5000, USA. [7]Laboratoire Interdisciplinaire Carnot de Bourgogne, UMR 6303 CNRS-Université Bourgogne Franche-Comté, 9 avenue Alain Savary, BP 47870, F-21078 Dijon Cedex, France. [8]Department of Chemical and Biomedical Engineering, FAMU-FSU College of Engineering, Florida State University, Tallahassee, FL 32310-6046, USA. [9]Present address: Oak Ridge National Laboratory, Nuclear Nonproliferation Division, Oak Ridge, TN 37831, USA. [10]These authors contributed equally: Bo Jiang, Jennifer Neu. ✉e-mail: kpage10@utk.edu; tsiegrist@fsu.edu

phonon picture in the literature, the direct observation of local bonding phenomena driving this phonon instability is lacking.

The difference between a displacive phase transition and an order-disorder transition has been discussed in different texts. Examples include the ferroelectric transition in $PbTiO_3$ which is mostly of displacive type with some local distortions, with soft phonon modes describing the transition[20–22]. In contrast, the ferroelectric transition in $BaTiO_3$ is considered order-disorder, with the titanium atom dynamically disordered along the [111] directions in the high-symmetry cubic phase[23–26]. The description of the phase transition in SnSe as a displacive transition is most often represented in the literature[3,4,7,10,11,13,16,27]. Thermodynamically, a displacive transition should be 2nd order with changes in the thermal behavior at the phase transition, whereas an order-disorder transition changes the configurational space, resulting in an entropy change. In the displacive transition model, atomic positions change gradually from low symmetry coordinates in the GeS-type structure (*Pnma*) to high symmetry coordinates in the TlI-type structure (*Cmcm*). The change in bonding around the Sn atom affects the band gap and could potentially create a partially filled band, where 2 Sn electrons are available for 4 orbitals. In addition, the Sn lone pair electrons are stereochemically active, producing an asymmetric bonding environment. The reduced band gap of the TlI-type structure, obtained by DFT calculations, is therefore due to the relationship between stronger bonding interactions and stronger charge transfer from Sn to Se. Enhanced phonon scattering decreases the lattice heat conductivity so that the power factor in SnSe reaches its highest value at temperatures near the phase transition (ZT of roughly 2.2–2.6 at 913 K)[4].

Several observations indicate the phase transition of SnSe may be unusual: primarily, all crystallographic studies performed using traditional Rietveld refinement methods result in larger displacement parameters for Sn than for Se[13,14]. Notably, the mass difference between Sn and Se is substantial and one would therefore expect that displacement parameters for Sn be smaller than for Se. The large displacement parameters of Sn and Se (based on a harmonic approximation) indicate that the Sn atom resides in a wide potential well. The double-well potential for the Sn atoms suggested recently indicates that the Sn atom can disorder into multiple positions[11], potentially resolves disparate observations of the system, as it is possible to overlay the GeS-type (*Pnma*) structure onto the TlI-type (*Cmcm*) structure, if a disordered (orientationally averaged) GeS-type (*Pnma*) based local model is used. In this case, the correlation length of the GeS-type atomic motif would be reduced as thermal excitations increase the disorder of the Sn atoms, which increasingly move between energetically accessible positions. A local probe is therefore needed to conclusively investigate if the local bonding in SnSe above the phase transition temperature retains the bonding observed in the GeS-type (*Pnma*) structure, or if there is a change in the orientation of the lone pair electrons as its average TlI-type structure type might suggest. The high real-space resolution intrinsic to time-of-flight (TOF) neutron pair distribution function (PDF) studies has proven key in elucidating local bonding distortions, nanoscale correlations, and other local to long-range structural complexities in a host of functional materials[25,28–31]. Herein we combine neutron TOF diffraction and PDF studies with ab initio molecular dynamics (AIMD) simulation techniques to investigate the short-range order in SnSe from room temperature to several hundred degrees above its crystallographic phase transition, thereby providing new insight into the curious nature of its phase transition.

## Results
### Average structure
A three-dimensional surface contour map with projection of neutron diffraction data measured as a function of temperature is shown in Fig. 1A. Specific diffraction peaks merge above 775 K, as highlighted in

Fig. 1B, a direct observation of the structural phase transition from GeS-type (*Pnma*) to TlI-type (*Cmcm*) SnSe. Rietveld refinement results by GSAS II[32] obtained at all temperatures are consistent with previous reports[13,14]. Refinements of two temperature points (400 K and 800 K) are shown in Fig. 1C. For comparison, the GeS-type (*Pnma*) structure model is applied up to 1000 K, and the TlI-type (*Cmcm*) structure model is applied from 750 K to 1000 K. The evolution of lattice parameters as well as atomic anisotropic displacement parameters (ADP) is shown in Supplementary Fig. 1, while the agreement factor ($R_w$), lattice parameters and atomic position parameters are provided in Table S1. A phase transition between 775 K and 800 K is clearly observed from the variation of lattice parameters, which correspond to the vanishing of (002) and (220) reflections in Figs. 1B and 2C. Furthermore, the temperature evolution of average Sn-Se bond distances derived from Rietveld fits, are shown in Fig. 1D (corresponding labels are indicated in (G)). The Sn-Se d1, d3, and d5 bond distances decrease in correlation with stronger bond strength and d2 and d4 increase in correlation with weaker bond strength. Overall, Sn-Se bonds become weaker at high temperature, resulting in underbonding and a pronounced softening at the phase transition. Additionally, thermal motions are underconstrained, resulting in strongly anharmonic "rattling"-type behavior. In this work and in previous Rietveld analysis studies, the application of a strict crystallographic model (with infinite correlation length) throughout the temperature range biases the analysis in favor of a static structure with large displacement factors. There is a consistent observation of larger displacement parameters for the Sn atoms with respect to the significantly lighter Se atoms within the low symmetry GeS-type (*Pnma*) structure, and persisting in the TlI-type (*Cmcm*) structure, as shown in Fig. 1E, F. Such behavior is often associated with disorder, where the heavy Sn atoms are displaced from their average position in the lattice, with static and/or dynamic disorder present.

### Neutron PDF and small-box modeling
Figure 2A shows a contour plot of PDFs measured as a function of temperature, with the corresponding 1D datasets aligned below as Fig. 2B. A number of important features can be directly observed in real space. First, significant changes in the real-space structure begin far below and continue gradually up to the crystallographic phase transition (for example, the first and second nearest-neighbor maxima below 3.75 Å merge into one broad asymmetric maximum between RT and 800 K, and a strong minimum emerges gradually at 25 Å as another disappears gradually at ~28 Å approaching ~675 K). These trends are juxtaposed with the trends in correlations between 3.75 Å and 12 Å, and specific higher *r* correlations (for example at ~15 Å and 24 Å) where no changes in position or intensity are observed as a function of temperature. Above the crystallographic phase transition, ~800 K, the only significant trend is a shift of all peaks to higher *r*, consistent with thermal expansion of the lattice.

A more in-depth interpretation can be made by modeling the local atomic structure of SnSe across the data series. Local (low-*r*) range fits to the 1.5–10 Å PDF region and medium (high-*r*) range fits to the 10–30 Å PDF region was initially fit in the small-box modeling program PDFgui[33] with the determined average structure models at each temperature (results summarized in Supplementary Figs. 2 and 3). The GeS type (*Pnma*) average structure model describes the medium-range structure well leading up to the crystallographic phase transition, and the TlI-type (*Cmcm*) average structure model does the same above the phase transition. Model parameters extracted from full-range PDF refinements (1.5 to 30 Å, Supplementary Fig. 4) provide close agreement to Rietveld refinement results. The story is different when fitting solely the low-*r* PDFs; select results are shown in Fig. 2C–E. The partial PDFs corresponding to the Sn-Sn, Sn-Se, and Se-Se pair-pair correlations extracted from small-box modeling are displayed below the data and fits. It is evident that the GeS-type (*Pnma*) model gives a high-quality fit to the PDF at 400 K in

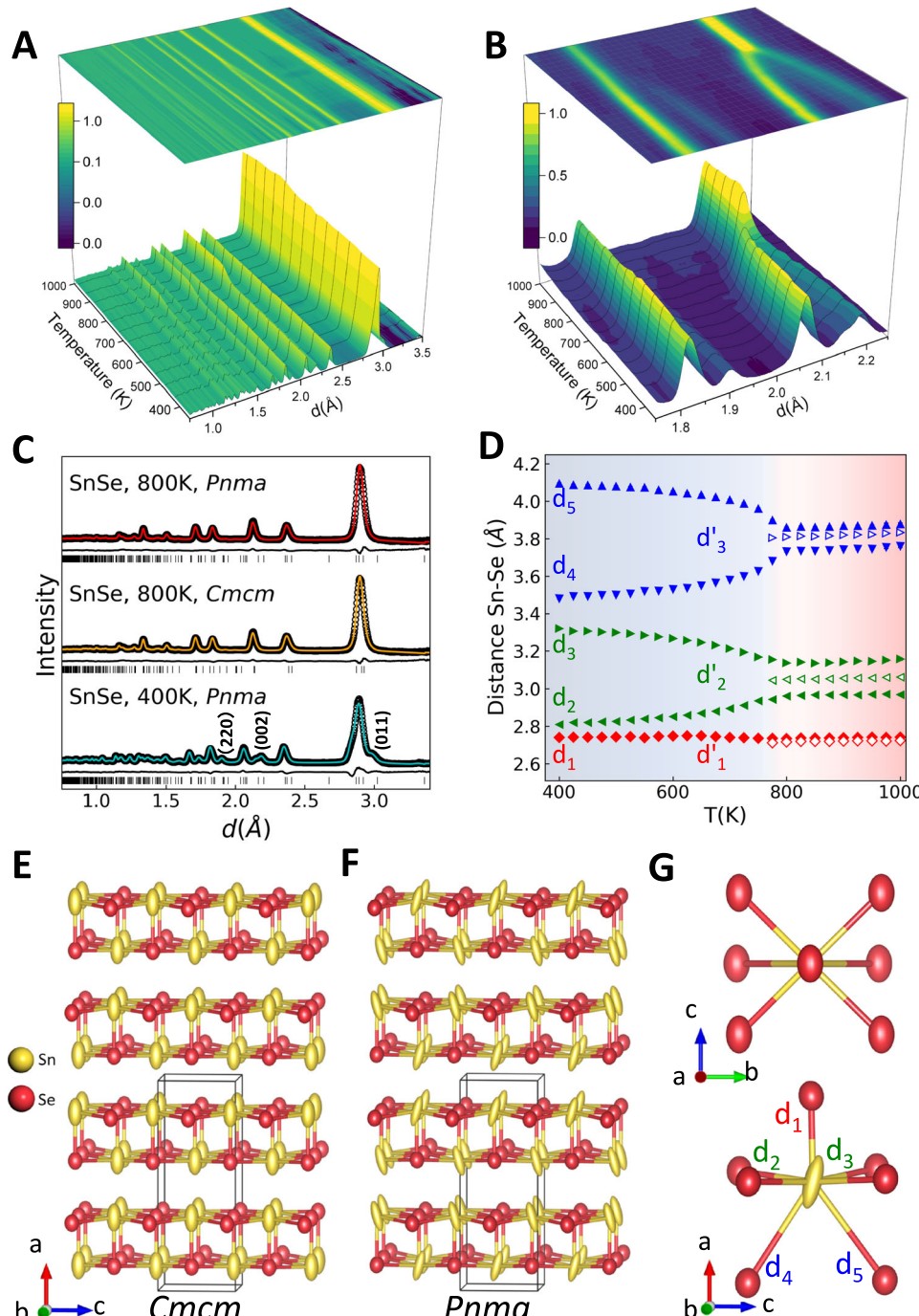

**Fig. 1 | Average structure of SnSe. A** 3D surface maps with projection of neutron diffraction data as a function of temperature, with **B** a close-up view of several peaks. **C** Results of Rietveld refinement of the data for 800 K (completed with both GeS-type (*Pnma*) and TlI-type (*Cmcm*) models) and 400 K (completed with GeS-type (*Pnma*) model), with data shown as black circles, model fits shown as red/orange/cyan lines, and difference curves shown as black lines (offset below data and fits). **D** Evolution of Sn-Se bond distances as a function of temperature from the Rietveld fits; values resulting from *Pnma* and *Cmcm* models are indicated with solid and hollow symbols, respectively. Error bars have been included but are smaller than the data symbols. The colored backgrounds indicate the boundary of the phase transition temperature. Rietveld refinement of the 800 K data in *Cmcm* space group yields the structure model shown in **E** compared with the *Pnma* refined structure model in **F**. First coordination polyhedron of SnSe (**G**) at 800 K using the *Pnma* model. Models employ 80% probability ellipsoids to emphasize the ADP obtained. Source data are provided as a Source Data file.

Fig. 2C. By contrast, the low-*r* PDF for 800 K is not well fit with the higher symmetry TlI-type (*Cmcm*) model, particularly the peak centered at 2.9 Å, corresponding to the Sn-Se nearest-neighbor correlation (Fig. 2D). The first and second Sn-Se correlations are changed significantly between 400 K and 800 K; notably the two distinct pair correlations merge into one asymmetric peak above 800 K. The persistent presence of this asymmetry confirms the need for a lower-symmetry model specifically symmetry model above the crystallographic phase transition. Thus, the GeS-type (*Pnma*) model was applied to the whole temperature range, resulting in similar quality fits at all temperatures (the result for 800 K is shown in Fig. 2E). The lower-symmetry model specifically provides a substantial improvement to the peak fit centered at 2.9 Å, with the refinement agreement factor $R_w$ dropping from 0.104 in Fig. 2D to 0.058 in Fig. 2E. The partial PDFs indicate this comes about

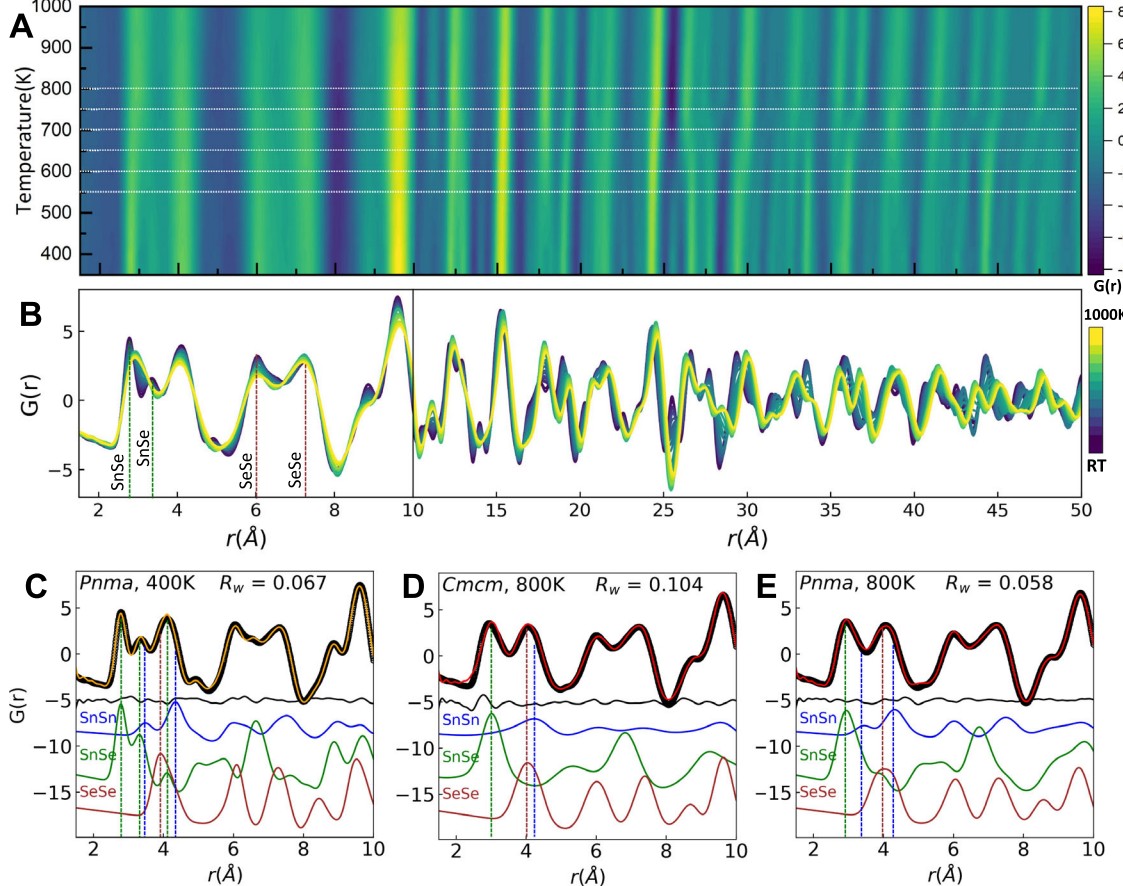

**Fig. 2 | PDFs and local structure in SnSe. A** The contour plot of neutron PDF data as a function of temperature for the real-space range of 1.5 to 50 Å (with data up to 10 Å shown on an expanded *r* scale). **B** The temperature-dependent PDF patterns are overlaid below for clarity. Experimental and simulated partial PDF calculated from small-box PDFgui fits for **C** 400 K using the GeS-type (*Pnma*) model, **D** 800 K using the TlI-type (*Cmcm*) model, and **E** 800 K using the GeS-type (*Pnma*) model. Source data are provided as a Source Data file.

through retained asymmetric bonding involving the Sn-Se nearest neighbors.

The temperature-dependent partial PDFs of the Sn-Sn, Sn-Se, and Se-Se correlations extracted from the local structure models are presented in Fig. 3A–C. It is clear that Sn-Sn correlations change little over the full temperature range, while Sn-Se correlations shift/merge significantly. A natural extension is to explore the length scale for this local ordering motif and the extent to which it survives above the crystallographic phase transition. Figure 3D compares the refinement agreement factor from variable *r* range modeling using the lower and higher symmetry variants of the SnSe structure for data collected at 600 K, 800 K, and 1000 K. Results for additional temperatures are summarized in Supplementary Figs. 5 and 6. Note that there are three correlation length scales: below ~12 Å, 12 Å to 24 Å, and above ~24 Å, corresponding to the approximately two unit cell lengths (2 × a lattice dimension) of the *Pnma*-like local order, a 'mixed phase' regime, and the *Cmcm*-like (average structure) order range. At 600 K (below the crystallographic phase transition), the lower-symmetry GeS-type (*Pnma*) model is the best description over the whole real-space range, meaning that the local structure matches the average structure. Above the transition, at 800 K and 1000 K, the GeS-type (*Pnma*) model also performs significantly better below ~12 Å and slightly better up to ~24 Å. Beyond that range, the high-temperature average TlI-type (*Cmcm*) model provides a similar quality of fit to the data. The detailed nature of Sn-Se bonding is found to be particularly sensitive to the length scale of the applied real-space analysis. Figure 3E displays the refined Sn-Se bond distances resulting from variable *r*-range PDF

fits for 800 K data. The distances within the layers ($d_1$, $d_2$, and $d_3$) remain nearly constant and equal to distances in the refined average structure with variation in the maximum real-space range probed ($r_{max}$). However, Sn-Se bonding to adjacent layers is found to be length-scale dependent, with $d_4$ decreasing and $d_5$ increasing as the $r_{max}$ (or probed correlation length scale) decreases. This is brought about by strong local Sn displacement, creating one shorter bond and one longer bond to Se atoms in adjacent layers. The average structure is thus realized through a superposition of the locally ordered states. These results demonstrate that SnSe maintains something of its low-temperature local symmetry across the whole temperature range, yet the coherence of these structural distortions decreases as temperature is increased. At the highest temperature probed here, 1000 K, this coherence approaches just 12 Å, or ~2 lattice lengths *a* of the SnSe structure. A technique that is sometimes applied to the analysis of Bragg diffraction data to give additional insight into phase transitions from lower-symmetry to higher-symmetry parent structures is the so-called symmetry-mode or distortion-mode Rietveld refinement[34]. Symmetry-mode analysis has recently been incorporated into a number of PDF data modeling workflows[35,36], and has been applied here in order to follow the length-scale dependence of the group-subgroup relationship between the *Cmcm* and *Pnma* structures. Results are shown in Fig. 2D, E and Supplementary Fig. 7. Four displacive mode amplitudes (*a*1, *a*2 for Sn and *a*3, *a*4 for Se, displayed in an inset) are applied here (within the TOPAS v6 software suite[37] in conjunction with models from ISODISTORT[38]) to model the changes of atom positions across the structural phase transition. The amplitude of displacements

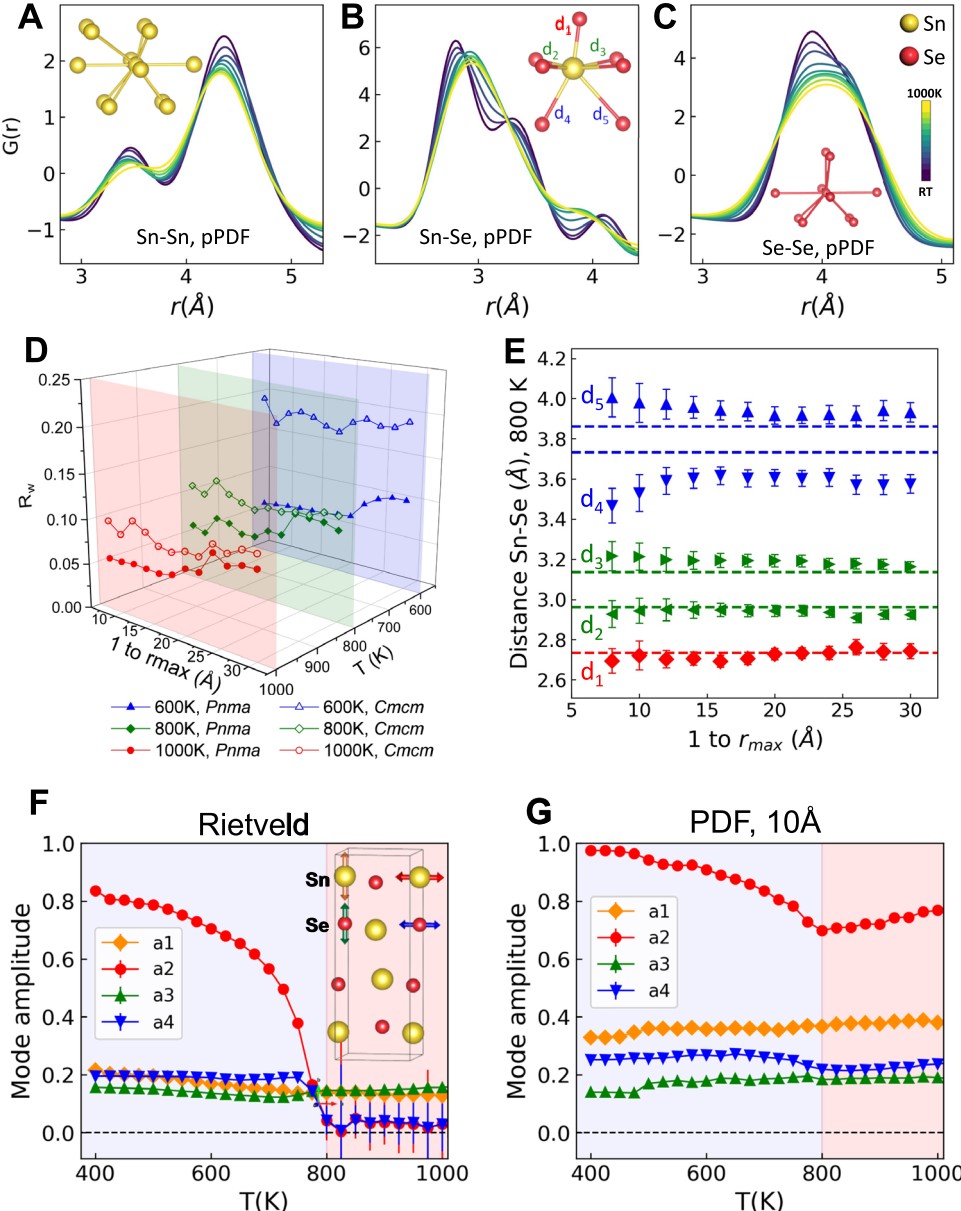

**Fig. 3 | Partial PDF and varying *r* range refinements.** Partial PDF resulting from GeS-type (*Pnma*) model fits for **A** Sn-Sn, **B** Sn-Se, and **C** Se-Se nearest-neighbor pairs as a function of temperature. **D** The agreement factor, $R_w$, resulting from small-box PDFgui analysis for GeS-type (*Pnma*) and TlI-type (*Cmcm*) models across varying $r_{max}$ range for select temperatures (600 K, 800 K, and 1000 K) in the data series. **E** Evolution of Sn-Se bond distances across varying $r_{max}$ range for 800 K data, with dashed lines showing the distances refined from Rietveld (GeS-type model)

analysis. Source data are provided as a Source Data file. Symmetry-mode-based **F** Rietveld and **G** low-*r* PDF (1.5 to 10 Å) refinements (with error bars) performed sequentially using the TOPAS v6 suite in conjunction with ISODISTORT. The colored arrows in the inset in **F** identifies the four modes in the TlI-type (*Cmcm*) structure fitted in the analysis. The colored backgrounds indicate the boundary of the phase transition temperature. If error bars are not visible they are smaller than the data markers.

*a*1 and *a*3 stay constant and low across the temperature range in both sequential Rietveld and *r*-dependent PDF refinements. By contrast, the *a*2 and *a*4 displacements show a sharp decline in their amplitudes at ~800 K, in good agreement with the ADP values from Rietveld refinement. Importantly, the amplitudes of the *a*2 and *a*4 modes retain elevated values for PDF refinements below 10 Å, in agreement with the maintained local symmetry discussed above.

A non-biased combinatorial appraisal of transition states (CATS) analysis[39,40] of changes in the data is shown in Supplementary Fig. 7, and it confirms the above assessment: the SnSe structure changes continuously and gradually from RT up to its crystallographic phase transition, and it does so in a non-uniform manner across real-space correlations (with pair correlations changing very little below 10 Å, and

with significant higher *r* changes peaking at the phase transition yet spanning the entire temperature regime). Further description of the CATS analysis is given in the Methods section. Overall, there are suggestions of both order-parameter-like transitions (gradual bonding changes as a function of T) and order-disorder type structural rearrangements (rigid/unchanging local ordering motifs that lose their coherence across the lattice with increasing T) at play in SnSe. The layer-type nature of the structure provides a mechanism for coherent Sn distortions within a layer, and reduced correlation/registry with neighboring layers. Se atoms will be affected as well, enabling easier disordering of the adjacent Sn atoms. The thermal energy allowing the Sn atom to disorder is therefore expected to manifest itself in structural changes well below the phase transition temperature. This is

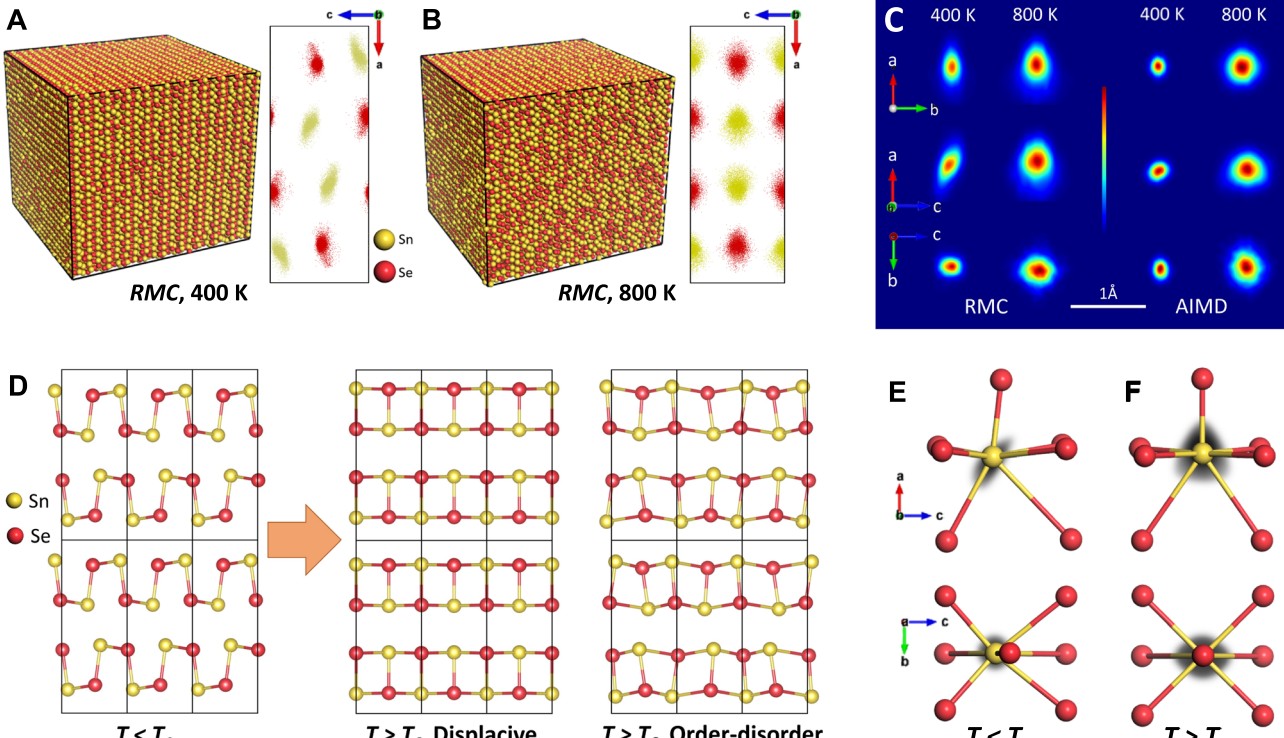

**Fig. 4 | Large-box and AIMD modeling.** Single RMC configurations after simulations for **A** 400 K and **B** 800 K, with corresponding RMC 'point cloud' models folded within a single unit cell on the right. **C** From top to bottom shows the temperature-dependent Sn probability density distributions projected onto the *ab*, *ac*, and *bc* plane from RMC and AIMD analyses. **D** Proposed displacive and order-disorder transition models in SnSe. The corresponding coordination polyhedron of **E** GeS-type (*Pnma*) and **F** TlI-type (*Cmcm*) structure is overlaid with Sn probability density distributions (in gray) from RMC analysis at 400 K and 800 K, representative of Sn atomic distributions at $T < T_C$ and $T > T_C$, respectively.

clearly apparent in Fig. 2 and Supplementary Fig. 7, where changes in longer distance correlations appear 100 K to 200 K below the nominal phase transition. The result across and above the SnSe phase transition is an average structure with higher observed symmetry and anomalously high ADP (consistent with the Rietveld analyses in our and others work). These phenomena are entirely consistent with the soft phonon view of the phase transition observed with inelastic neutron scattering and supported by theoretical work.

## Large-box and AIMD modeling

We pursued large box Reverse Monte Carlo (RMC) modeling using RMCProfile[41] and AIMD simulations to further investigate the nature of the large ADP and the origins of local-to-long range complexity in SnSe structures. The RMC modeling was completed on 400 K and 800 K data with a supercell >100 Å on each side, as shown in Fig. 4A, B, with corresponding folded unit cell 'point clouds' on the right. Excellent fits for both temperatures were obtained for the whole PDF range, as shown in Supplementary Fig. 9, with fits to the normalized structure factor, F(Q), and Bragg diffraction data shown in Supplementary Fig. 10. We further performed AIMD simulations for theoretical support (see Supplementary Fig. 11). The radial distribution function (RDF) for all pair-pair correlations obtained from AIMD simulations are in good agreement with RDFs calculated from the RMC models (Supplementary Fig. 11). ADPs resulting from Rietveld analysis are compared with the atomic 'point clouds' and trajectories from RMC analysis and AIMD simulations in Supplementary Fig. 12, respectively, for (a) 800 K and (b) 400 K datasets. Note that the shape and size of 'point clouds' resulting from RMC modeling are comparable to the shape and size of ADP thermal ellipsoids from Rietveld refinement. The RMC 'point clouds' of Sn and Se exhibit dispersive quasi-prolate ellipsoid distributions, similar to the ADP distributions. However,

the density plots from RMC and AIMD analyses indicate Se atom sites that are small ellipsoids (see Supplementary Fig. 12), and Sn atom sites that are larger ellipsoids asymmetric in nature (Fig. 4C, E, F).

In divalent Sn, the non-zero overlap of the 5 s and 5p orbitals tends to produce a bonding asymmetry, which is usually ascribed to a stereochemically active lone pair. The bonding asymmetry results in three shorter bonds that tend to be nearly orthogonal, and three or more longer bonds, that are also affected by the orientation of the lone pair electrons. The asymmetry in the Sn atom locations in SnSe is shown in more detail in Fig. 4C, where the distributions are projected along different unit cell directions. It is clear at low temperature, via ADP, RMC, and AIMD analyses, that neighboring Sn atoms distort/rotate to avoid end-to-end bonding, resulting in coherent/cooperative tilting of their locations. At higher temperatures, the Sn atoms continue to be off-center, retaining their local bonding configurations, but in such a way that the distributions are no longer coherent (they do not displace in the same directions in all unit cells). The Sn probability distribution grows in the (ab)-plane at high temperature, retaining a similar shape. However, a triangle shape Sn probability distribution is formed at high temperature in the (ac)-plane, with Sn atoms distorting/rotating in both directions. The disorder of the Sn atoms affects the coordinating Se atoms and is expected to change the form of the local minima. Thus, the existence of orientationally averaged local distortions in the SnSe system is confirmed leading up to and persisting above its global crystallographic transition. An illustrated comparison of the displacive and order-disorder phase transition models is given in Fig. 4D, with experimentally determined (from RMC analysis) probability density distributions of Sn (in gray) overlaid with average (from Rietveld analysis) first coordination polyhedron in Fig. 4E, F. The observed long-range structure at high temperature derives from locally off-centered motifs that are orientationally averaged, similar to the dynamic

order-disorder phase transition observed in other Jahn-Teller and lone pair-driven phase transition phenomena[42]. It should be noted that order-disorder behavior need not rule out a displacive phase transition; its signatures are observed during the displacive-type phase transitions of the perovskite $PbTiO_3$[43] and even in cases where a crystallographic phase transition is absent upon warming, such as in the cubic rock-salt PbTe[44]. In the latter case, an unusually correlated local structure dipole formation linked to coupled soft optical and acoustic modes were proposed[45]. The order-disorder nature of SnSe is similarly important to its electrical and thermal transport properties.

It is instructive to consider several limitations of the present study. First, the neutron total scattering method used in this work is an energy-integrated scattering approach, meaning that the data incorporates pair correlations from elastic (long-lifetime) correlations as well as from dynamic correlations. Future experimental work with the dynamic PDF technique (completed with an inelastic scattering instrument)[46,47] could discriminate the energy dependence of the high-temperature local atomic distortions uncovered. Remarkably large and anisotropic dynamics have recently been uncovered and linked to the contradictory thermal and electronic conductivity properties in the cubic thermoelectric GeTe through energy-resolved PDF techniques[48]. However, such an endeavor is challenged as present-day inelastic instruments with the requisite energy resolution and energy range lack the $Q$-range required for high-quality real-space PDF. Second, as a powder-averaged technique, the orientational dependence of local atomic correlations is lost. Future experimental work with 3D-PDF methods[45,49] on single crystal SnSe could further clarify the present work. To be clear, the present work establishes persisting local Sn dipoles, but cannot differentiate whether they are fluctuating/dynamic or static and orientationally averaged, nor in what crystallographic directions and to what extent the Sn ions propagate and fluctuate in time.

This work affirms that a model of the phase transition in SnSe must include the following characteristics:

(i)    The Sn-Se local structure remains asymmetric at all temperatures
(ii)   The correlation length over which one Sn position affects the adjacent Sn position becomes shorter with increasing temperature
(iii)  The potential energy surface of the Sn is likely a multi-minima surface, where additional energy minima become accessible with increasing temperature
(iv)   At a high enough temperature, the Sn atoms are likely dynamically disordered over multiple local minima positions, wherein the average structure is consistent with the TlI-type motif.

We further note that observations of such a dynamic and/or orientationally averaged atomic configuration will be biased by the specific length- and time-scale sensitivities of the employed probes. The nature uncovered here in SnSe through examination of its local to long-range structural motifs thus fills in key details among previous work addressing its phase transition.

In summary, we find that the structural phase transition in SnSe features an asymmetric Sn-Se coordination environment consistent with the GeS-type structure at all temperatures probed. Above the crystallographic phase transition, thermal excitations allow the Sn atom to access alternate higher energy positions, reducing the length scale over which the locally off-centered Sn atoms are correlated. The dynamic disorder destroys the long-range coherence, allowing a satisfactory description of the average atomic arrangement in the TlI-type structure, albeit with large anisotropic ADPs for Sn. Additionally, indications of the phase transition are already apparent at temperatures well below 800 K. The correlation length is found to be strongly temperature dependent and diverges at the phase transition temperature. The dynamic disorder, described as "rattling" of the Sn atoms, creates a strong phonon scattering cross section that couples efficiently to the phonon spectrum increasing the backscattering rate,

and thus decreasing the lattice thermal conductivity. The change in correlation length of the local Sn motif observed via neutron total scattering reconciles previously juxtaposed views of the phase transition in thermoelectric SnSe.

## Methods

### Materials synthesis

Polycrystalline tin selenide (SnSe) was synthesized by solid-state reaction. The synthesis method employed follows methods outlined in the literature[50–52]. In detail, fused silica tubes containing a 1:1 stoichiometric ratio of metallic tin and selenium shot were sealed under vacuum. Throughout the sealing process, the temperature of the outside surface of the silica tubes was controlled to limit sublimation of selenium prior to sealing. The ampoules containing 5 g charges were placed in a muffle furnace and heated from room temperature to 510 K over 110 h. Then, ampoules were removed from the furnace, shaken by hand in order to break up solids to assist the uniform reaction of selenium with tin, and returned to the furnace. The temperature was increased to 775 K over 12 h and then finally increased to 1175 K over 60 h, followed by cooling to room temperature in the furnace. Samples obtained were large silver-gray chunks. The sample was ground with an agate mortar and pestle for initial phase analysis by powder X-ray diffraction, carried out in a standard Bragg-Brentano geometry (Scintag PAD-V) powder X-ray diffractometer equipped with $CuK_\alpha$ radiation and graphite diffracted beam analyzer. Rietveld refinement of the powder X-ray pattern confirmed phase purity of samples obtained.

### Neutron experiment and average structure analysis

The polycrystalline samples were loaded into vanadium sample holders under an inert atmosphere, and placed in the furnace at the NOMAD time-of-flight (TOF) instrument at the ORNL Spallation Neutron Source (SNS). Neutron powder PDF data was reduced to a Q-range of 27 Å$^{-1}$ to ensure that the local distortions are well represented in real-space. Datasets were collected for different temperatures, from ambient temperature to 1000 K, at intervals of 25 K (we use only 400 K to 1000 K in manuscript due to relatively poor temperature control at furnace temperatures below 400 K). Standard background subtraction and normalization routines and conversion from total scattering to PDF were performed using the Addie software suite[53]. We have selected the standard space group setting *Pnma* for the orthorhombic GeS-type structure; in order to directly relate the unit cell parameters to those of the TlI-type structure, space group setting *Bbmm* is used. We note that *Cmcm* is the standard setting of *Bbmm*, and we refer to the structure as *Cmcm* herein, and within the main paper body. Rietveld refinements were carried out using the General Structure Analysis System (GSAS-II)[32] with scale factor, zero shift, background, unit cell parameters, peak profile parameters, and anisotropic temperature factors refined. Data refinements in GSAS-II were checked for stability and the presence of local minima by randomly changing parameters and restarting the refinements.

### Pair distribution function (PDF) modeling

The PDF method, as a powerful total scattering (Bragg and diffuse scattering) technique, has been widely used to examine both the local structure (diffuse component) and long-range structure (Bragg component) of disordered crystalline materials. The PDF G(r) describes the probability of finding an atom at a radial distance of $r$ from a given atom[54], as $G(r) = 4\pi[\rho(r)-\rho_0]$, where $\rho(r)$ and $\rho_0$ are the local and average atomic number density, respectively. Sine Fourier transformation of the normalized total structure function S(Q) were applied to obtain the G(r):

$$G(r) = \frac{2}{\pi} \int_0^\infty Q[S(Q) - 1] \sin(Qr)dQ \qquad (1)$$

where $Q$ equals $4\pi sin\theta/\lambda$. Temperature-dependent trends were analyzed in a model-independent fashion using CATS analysis[39], where the endmembers selected were datasets at 400 K and 1000 K. In brief, the approach solves for the best combination of the first and last datasets (as fractions of unity) that best match each intermittent dataset, reporting a phase fraction (phi) of the first dataset and a residual value of each combined fit for each dataset. To calculate the distance between two datasets, we employed a convolutional comparison metric[40] to mitigate the effect of peak shifts attributed to thermal expansion on the unit cell. The $r$-dependent CATS was performed with a constant box-width of 4 Å from ranges of $r = 1$ Å $-5$ Å, up to $r = 46$ Å $-50$ Å, sliding 1 Å at a time (45 different boxes). The results for phi in Supplementary Fig. 8A, show that the PDFs are changing gradually leading up to the crystallographic phase transition, and then minimally after that point. Concurrently, the residual map, shown in Supplementary Fig. 8B, follows a pattern of gradually increasing, peaking at the crystallographic phase transition, and quickly fading away. The map reveals very little residual below approximately $r = 10$ Å, indicating a common low-$r$ structure across the entire temperature window.

PDF fitting was performed both by 'small-box' modeling using PDFgui[33,37] and 'large-box' modeling by RMC simulations with the RMCprofile[41] software. The initial structures obtained from Rietveld refinement were used in PDFgui modeling with the following parameters refined: scale factor, correlated atomic motion (delta2 parameter for low-$r$ regions), unit cell parameters, atomic positions, and isotropic/anisotropic atomic displacements. The refinement agreement factor $R_w$[54] are defined as:

$$R_w = \left[\frac{\sum w_i \left(G_i^{exp} - G_i^{calc}\right)^2}{\sum w_i \left(G_i^{exp}\right)^2}\right]^{0.5} \qquad (2)$$

The RMC technique using total scattering data has been previously used to explore structural phase transitions of crystalline materials[55–57]. It is worth noting that RMC has a capacity for models containing more than ten thousand atoms, enabling one to fit the experimental total scattering data in real and reciprocal space simultaneously. Here, RMC analysis was completed using a $10 \times 25 \times 25$ supercell (50000 atoms) from both the initial average structure of GeS ($Pnma$) and TlI-type ($Bbmm$, standard $Cmcm$) unit cell. Finally, the RMC model was obtained by fitting PDF D(r)[58], scattering function F(Q), and the neutron diffraction patterns simultaneously. In the RMC simulations distance-windows restrictions were applied to avoid unphysical bond lengths. The simulations ran for more than 2500 min on one 3.2 GHz core CPU, generating ~0.5–1.0 × 10^7 moves. A total of 5 final RMC configurations for each selected temperature were produced to combine the results for further analysis.

## AIMD simulations

To investigate the finite temperature of atomic and lattice dynamic in SnSe, we performed AIMD using density functional theory (DFT) with the Vienna Ab initio Simulation Package (VASP) code[59,60] using the *PBEsol* functional[61]. The NVT[62] ensemble was carried out via the algorithm of Nosé[63] to control the temperature oscillations during the DFT calculations. Brillouin zone integration was done on a single gamma-centered k-point and the cutoff energy was reduced to 300 eV for all AIMD calculations. In this work, $2 \times 4 \times 4$ supercells ($Sn_{128}Se_{128}$, 256 atoms) were used for GeS ($Pnma$) and TlI-type ($Bbmm$, standard $Cmcm$) structures. The lattice parameters at 400 K and 800 K from Rietveld refinement are adopted for the AIMD simulations at 400 K and 800 K. Each AIMD simulation was run for 5000 steps with a time-step of 5 fs. The last 3000 steps were used to generate the time-averaged structure at each temperature.

## Reporting summary

Further information on research design is available in the Nature Portfolio Reporting Summary linked to this article.

## Data availability

The data supporting the findings of this study are available from the corresponding author on request. Source data are provided with this paper.

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

## Acknowledgements

Structure analysis, including work by B.J., D.O., and K.P. was primarily supported by the Basic Energy Sciences Office of Science Early Career

Award: Exploiting Small Signatures: Quantifying Nanoscale Structure and Behavior. This work was supported by the "investments for the future" project ISITE-BFC (contract ANR-15-IDEX-0003). The computing resources were made available through the VirtuES project as well as the Compute and Data Environment for Science (CADES) at the Oak Ridge National Laboratory, which is supported by the Office of Science of the U.S. Department of Energy under Contract no. DE-AC05-00OR22725. This research made use of the NOMAD instrument at the Spallation Neutron Source, a DOE Office of Science User Facility operated by the Oak Ridge National Laboratory. J.N. and T.S. acknowledge support from the National Science Foundation, grant DMR-1606952. Work by J.N. and T.S. was carried out at the NHMFL, which is supported by the National Science Foundation under grant DMR-1644779 and the State of Florida.

## Author contributions

T.S. planned the project with K.P., and J.N. prepared materials and performed the neutron diffraction analysis. D.O., K.P., and S.A.J.K. performed the neutron total scattering experiments. B.J. and K.P. performed the neutron pair distribution function analysis and theory calculations analysis. D.O. and K.P. performed the CATS analysis. B.J. and J.N. wrote the manuscript under the guidance of K.P. and T.S.

## Competing interests

The authors declare no competing interests.
