## [Peer Review File · Nature Communications]

The Curious Case of the Structural Phase Transition in SnSe Insights from Neutron Total ScatteringThis manuscript has been previously reviewed at another journal that is not operating a transparent peer review scheme. This document only contains reviewer comments and rebuttal letters for versions considered at *Nature Communications*.

REVIEWER COMMENTS

Reviewer #1 (Remarks to the Author):

The authors well addressed my technical questions, I personally think that this current manuscript can be published in Nature Communications.

Reviewer #2 (Remarks to the Author):

In this manuscript, the authors demonstrate a detailed analysis of local as well global structures of SnSe over wide range of temperature (well above the structural phase transition). They establish an order-disorder type transition where the characteristics of asymmetric Sn-Se coordination remains consistent with the GeS-type structure throughout the temperature range. Beyond the phase transition, thermal energy enables the Sn atom to access other possible higher energy positions decreasing the length scale over which the locally off-centered Sn atoms are correlated. Moreover, the phase transition possibly occurs well below the temperature 800 K evident from different indications. The correlation length is deeply temperature dependent and diverges at the phase transition temperature. This rattling behaviour of Sn atoms generate the soft optical modes which enhances the scattering rate of heat carrying phonons, and thus suppresses the lattice thermal conductivity. All the neutron PDF data are well analysed which provides a nice platform to understand the insights of the type of phase transition in SnSe at high temperature and its consequences on thermal transport. The manuscript can be publishable after addressing the following concerns.

The rattling behaviour is generally observed for those atoms which are very weakly bonded, however, Sn on the other hand in SnSe is strongly bonded with some of the Se for all over the studied temperature. Therefore, how does the author justify the terminology 'rattling' in this present scenario? In Figure 3c, the green colour data looks splitted for the characteristics of Se-Se local bonding environment? Why is it so and what does this signify? Does the author have any comments on the bond strength of Sn-Se at higher temperature compared to low temperature. In addition to that, authors must mention the correlation length in the PDF analysis and include the standard deviations in Fig 3f,g and S1a-b.

Liberation of Se from SnSe is very common factor when it comes to the high temperature. Thus, how do the authors ensure about their results at 1000 K? The authors claims that the reduced band gap of the TII-type structure of SnSe obtained by DFT calculations is due to charge transfer from Sn to Se. Isn't it due to the stronger bonding interaction between Sn and Se or the maybe be the reduced bond length at higher temperature? And at last, how does the order-disorder nature of SnSe affect the electronic transport as claimed by the authors.

Reviewer #3 (Remarks to the Author):

This manuscript reports a total scattering study of the high temperature phase transition in SnSe, an important thermoelectric. I reviewed this manuscript when it was submitted to Nature Materials (I was reviewer 3). In my report I recommended rejection for two key reasons: (i) the data and their analysis did not support the conclusions drawn, and (ii) that the new data – as valuable as they will be to the community – did not highlight the need for any fundamental shift in understanding of this phase transition. A large number of technical points were raised – by me and by the other referees.

The manuscript has now been revised and submitted to Nature Communications. Scientifically speaking, this is a much better paper, and the authors have done a good job of incorporating the various changes suggested by the referees at Nature Materials. My objection (i) above is largely addressed. Point (ii) still stands, however. The story that emerges in this paper is entirely consistent with the established understanding of this phase transition in SnSe. This is a valuable study that should be reported, but it simply does not involve a conceptual advance or sufficiently surprising result that would justify publication in a higher-impact journal or capture the

imagination of a broader audience. Plenty of similar-level studies have been published in well-respected community journals, and I think those are the right place for this study too. This point was flagged by all referees at the previous round of review (nb with similar journals suggested). The response from the authors contained no substantive rebuttal to this point.

In summary: this is a good (now scientifically correct) study that deserves to be published but I am unconvinced that any of the nature journals (e.g.) is the right place. I encourage the authors to submit to PRB, PRM, or Chem Mater so that the relevant community can engage with their nice work as soon as possible.

RESPONSE TO REVIEWERS' COMMENTS

We wish to express our gratitude for the helpful questions and constructive comments raised by the Reviewers concerning our manuscript NCOMMS-22-49355 under consideration at *Nature Communications*. We have carefully studied the Reviewers' comments (laid out below in italicized font) and revised our manuscript accordingly. All new and changed text is shown in our response and in a highlighted version of our manuscript with **yellow marking**, while the deleted text is shown with **red-strike** through the font. Our detailed response to each of the Reviewers' comments is given below.

Reviewer #1 (Remarks to the Author):

The authors well addressed my technical questions, I personally think that this current manuscript can be published in Nature Communications

Response to Reviewer #1's comments: We are very grateful to the referee for the careful reading of the paper and their detailed suggestions regarding our previous version which helped us to improve it considerably.

Reviewer #2 (Remarks to the Author):

In this manuscript, the authors demonstrate a detailed analysis of local as well global structures of SnSe over wide range of temperature (well above the structural phase transition). They establish an order-disorder type transition where the characteristics of asymmetric Sn-Se coordination remains consistent with the GeS-type structure throughout the temperature range. Beyond the phase transition, thermal energy enables the Sn atom to access other possible higher energy positions decreasing the length scale over which the locally off-centered Sn atoms are correlated. Moreover, the phase transition possibly occurs well below the temperature 800 K evident from different indications. The correlation length is deeply temperature dependent and diverges at the phase transition temperature. This rattling behaviour of Sn atoms generate the soft optical modes which enhances the scattering rate of heat carrying phonons, and thus suppresses the lattice thermal conductivity. All the neutron PDF data are well analysed which provides a nice platform to understand the insights of the type of phase transition in SnSe at high temperature and its consequences on thermal transport. The manuscript can be publishable after addressing the following concerns.

Response to Reviewer #2's comments: We are grateful for the reviewer's professional review of our article. Detailed responses and corrections are listed below each of the review comments provided.

The rattling behaviour is generally observed for those atoms which are very weakly bonded, however, Sn on the other hand in SnSe is strongly bonded with some of the Se for all over the studied temperature. Therefore, how does the author justify the terminology 'rattling' in this present scenario?

Response to Reviewer #2's comments: Rattling is a generic term for underconstrained structural elements, originally used for atoms loosely bound in highly symmetric 3D potentials (as in skutterudite, clathrates etc). It can also be used in other geometries, for example 1D disorder of atoms split between two sites, or indeed in SnSe, where the Sn atom oscillates between off-center positions. The key point here is that 'rattling' in all cases refers to situations where the atom in question spends almost no time in the time-average central position. We therefore feel justified in using it here. Often "rattling" is considered random, whereas in our case, there are positions that are more likely than others. We have used the term rattling, as the Sn atom is found in different positions consistent with the GeS-type structure, rattling dynamically.

In Figure 3c, the green colour data looks splitted for the characteristics of Se-Se local bonding environment? Why is it so and what does this signify?

Response to Reviewer #2's comments: As the reviewer states, the Se-Se nearest-neighbor pair correlation exhibits multiple peaks. The Se is situated in a ten-coordinated polyhedron surrounded by Se atoms. It features four distinct bond lengths between 3.85 Å and 4.4 Å, as illustrated in the inset model in Figure 3c.

Does the author have any comments on the bond strength of Sn-Se at higher temperature compared to low temperature.

Response to Reviewer #2's comments: The strength of a bond in a crystal structure is related to its length, with ions at shorter distance forming a stronger bond (<https://pubs.acs.org/doi/full/10.1021/acs.inorgchem.2c02766>). As seen in Figure 1D, the Sn-Se d1 to d5 bond distances vary as temperature increases. d1, d3, and d5 decrease (in correlation with stronger bonding) and d2 and d4 increase (in correlation with weaker bonding). Above 800K, the strength of all Sn-Se bonds becomes weaker as bond distances increase. In a band structure picture of the IV-VI materials, there is a delicate interplay between partly-filled conduction bands and high-symmetry structures stabilized by resonance effects. In a simple chemical picture, this reduces to competition between lone-pair formation, and electron delocalization. The key point is that electron counting reveals that these structures are 'underbonded' at high temperature. In the case of GeTe for example, each bond has a formal order of only 0.5 in the high temperature cubic phase. This results in a pronounced softening at the phase transitions of these materials, and thermal motions are underconstrained, resulting in strongly anharmonic rattling-type behavior. We have added a note regarding this point to the discussion of Figure 1D in the manuscript in Page 3: "The Sn-Se d1, d3, and d5 bond distances decrease in correlation with stronger bond strength and d2 and d4 increase in correlation with weaker bond strength. Overall, Sn-Se bonds become weaker at high temperature, resulting in underbonding and a pronounced softening at the phase transition. Additionally, thermal motions are underconstrained, resulting in strongly anharmonic "rattling"-type behavior."

In addition to that, authors must mention the correlation length in the PDF analysis and include the standard deviations in Fig 3f,g and S1a-b.

Response to Reviewer #2's comments: We included a description of the correlation length scale in the caption of Figure S6. We have moved this to the main manuscript, "Note that there are three correlation length scales: below ~12 Å, 12 Å to 24 Å, and above ~24 Å, corresponding to the approximately two unit cell lengths ($2 \times a$ lattice dimension) of the *Pnma*-like local order, a 'mixed phase' regime, and the *Cmcm*-like (average structure) order range." Additionally, we have checked all figures and added the standard deviations (error bars) where they were missing. Some of the error bars are smaller than the data markers, so are thus invisible in the Figures.

Liberation of Se from SnSe is very common factor when it comes to the high temperature. Thus, how do the authors ensure about their results at 1000 K?

Response to Reviewer #2's comments: The liberation of Se from SnSe can occur due to weaker bonds between the Sn and Se atoms at high temperatures. The extent of Se liberation at high temperatures will depend on several factors. In order to investigate the effect of Se liberation at high temperatures, the lattice parameters were refined from neutron diffraction data as the temperature was increased to 1000 K and then decreased back to room temperature, as shown below. Our neutron diffraction analysis did not reveal the presence of any obvious Se vacancies. It is worth mentioning that previous literature (*PHYSICAL REVIEW B* 99, 035306 (2019)) suggests that the SnSe prefers a Se-rich composition at low temperatures and a Se-poor composition at higher temperatures, and the very low concentration ($< 10^{17}/\text{cm}^3$) of V_{Se} and $V_{\text{Sn}} + nV_{\text{Se}}$ becomes saturated above 800 K. In conclusion, although the tested temperatures in our work are as high as 1000 K, the impact of possible Se liberation has been ignored as our work mainly focuses on the analysis of data below 800 K.

Figure 1. The lattice parameters refined from neutron diffraction patterns show a similar temperature dependence upon both increasing and decreasing temperature. No obvious Se vacancies are observed from the Rietveld refinement results.

The authors claims that the reduced band gap of the TII-type structure of SnSe obtained by DFT calculations is due to charge transfer from Sn to Se. Isn't it due to the stronger bonding interaction between Sn and Se or the maybe be the reduced bond length at higher temperature?

Response to Reviewer #2's comments: We believe multiple explanations for the reduced band gap are reasonable. The reduced band gap generally indicates a stronger charge transfer between Sn and Se (corresponding to stronger bonding interactions), which can lead to increased electrical conductivity and other properties. The DFT calculations are performed at zero K, so it is challenging to account for the effects of temperature on bond length without more sophisticated GW methods (*Scientific Reports volume 6, Article number: 26193 (2016)*). We have added a note indicating **the relationship between stronger bonding interactions and stronger** charge transfer to the manuscript in Page 2.

And at last, how does the order-disorder nature of SnSe affect the electronic transport as claimed by the authors?

Response to Reviewer #2's comments: As the reviewer notes above, the rattling behavior of Sn atoms generates soft optical modes which enhances the scattering rate of heat carrying phonons, suppressing the lattice thermal conductivity and impacting thermal transport. The order-disorder nature of SnSe has a direct impact on electronic transport properties which determine the thermoelectric performance. The ordered state exhibits higher electrical conductivity compared to the disordered state which results in reduced electronic mobility and increased electronic scattering. Therefore, understanding and controlling the order-disorder nature of SnSe is crucial for optimizing its electronic properties for various applications such as thermoelectric, photovoltaics, and other electronic devices.

Reviewer #3 (Remarks to the Author):

This manuscript reports a total scattering study of the high temperature phase transition in SnSe, an important thermoelectric. I reviewed this manuscript when it was submitted to Nature Materials (I was reviewer 3). In my report I recommended rejection for two key reasons: (i) the data and their analysis did not support the conclusions drawn, and (ii) that the new data – as valuable as they will be to the community – did not highlight the need for any fundamental shift in understanding of this phase transition. A large number of technical points were raised – by me and by the other referees.

The manuscript has now been revised and submitted to Nature Communications. Scientifically speaking, this is a much better paper, and the authors have done a good job of incorporating the various changes suggested by the referees at Nature Materials. My objection (i) above is largely addressed. Point (ii) still stands, however. The story that emerges in this paper is entirely consistent with the established understanding of this phase transition in SnSe. This is a valuable study that should be reported, but it simply does not involve a conceptual advance or sufficiently surprising result that would justify publication in a higher-impact journal or capture the imagination of a broader audience. Plenty of similar-level studies have been published in well-respected community journals, and I think those are the right place for this study too. This point was flagged by all referees at the previous round of review (nb with similar journals suggested). The response from the authors contained no substantive rebuttal to this point.

In summary: this is a good (now scientifically correct) study that deserves to be published but I am unconvinced that any of the nature journals (e.g.) is the right place. I encourage the authors to submit to PRB, PRM, or Chem Mater so that the relevant community can engage with their nice work as soon as possible.

Response to Reviewer #3's comments: We thank again the reviewer for the time and effort in reviewing the previous version and current version of our manuscript. We are pleased the reviewer positively assesses the technical improvements to our work. We believe what we have presented in our paper aligns with NC's aim, which is to "showcase significant advancements that are relevant to specialists" (<http://nature.com/ncomms/aims>). We also believe our work will motivate further studies in the thermoelectric field, specifically on SnSe. One of the authors of our article, Dr. Simon Kimber, has just published an article on the use of energy-resolved variable-shutter PDF technique to study the static disorder and atomic motions in the cubic thermoelectric GeTe in *Nature Materials* (<https://doi.org/10.1038/s41563-023-01483-7>). Such a study would on SnSe would be a natural extension of the present work. We also note since submitting our revision a paper with similar scope has been submitted on the arXiv: <https://arxiv.org/abs/2301.05773>, with the topic of characterization of complex local Sn off-centering across the phase transition in SnSe using x-ray PDF methods. In comparison, our paper provides more detailed atomic modeling of the dynamic order-disorder nature discovered in SnSe by examining its local to long-range structural motifs.

Finally, a few additional minor edits have been completed:

- The order of all author affiliations has been adjusted.
- We added error bars to Figure 3F and Figure S4.
- In Figure 4C, the correction has been made from "300K" to "400K" for AIMD analyses.
- We have corrected the color gradient in Figure 3(a)(b)(c) to make it consistent with the color bar.
- The colors of the Sn and Se atoms in the insert figure of Figure 3F and Figure S7 have been corrected.

To summarize, we sincerely thank all reviewers for the valuable feedback, and we have improved the quality of our manuscript according to the reviewer's suggestions. We have also amended the error we discovered after submission as described above.

Sincerely yours,

Katharine Page
Assistant Professor
Materials Science and Engineering Department
University of Tennessee

REVIEWERS' COMMENTS

Reviewer #2 (Remarks to the Author):

I have gone through the revised manuscript. Author have answered all the questions satisfactorily. The paper can be accepted as is.

RESPONSE TO REVIEWERS' COMMENTS

Reviewer #2 (Remarks to the Author):

I have gone through the revised manuscript. Author have answered all the questions satisfactorily. The paper can be accepted as is.

Response to Reviewer #1's comments: We appreciate the referee's valuable feedback and suggestions that helped us improve the quality of the manuscript.

Response to Reviewer #2's comments:

To summarize, we sincerely thank all reviewers for the positive feedback, and thanks for your time and effort in reviewing our work.

Sincerely yours,

Katharine Page
Assistant Professor
Materials Science and Engineering Department
University of Tennessee